# Optimization of Samples for Remote Sensing Estimation of Forest Aboveground Biomass at the Regional Scale

**Qingtai Shu [1], Lei Xi [1], Keren Wang [2], Fuming Xie [3], Yong Pang [4,*] and Hanyue Song [1]**

[1] College of Forestry, Southwest Forestry University, Kunming 650224, China
[2] Institute of Highland Forest Science, Chinese Academy of Forestry, Kunming 650224, China
[3] Institute of International Rivers and Eco-Security, Yunnan University, Kunming 650500, China
[4] Institute of Forest Resource Information Techniques, Chinese Academy of Forestry, Beijing 100091, China
[*] Correspondence: pangy@ifrit.ac.cn; Tel.: +86-135-2158-8630

**Abstract:** Accurately estimating forest aboveground biomass (AGB) based on remote sensing (RS) images at the regional level is challenging due to the uncertainty of the modeling sample size. In this study, a new optimizing method for the samples was suggested by integrating variance function in Geostatistics and value coefficient (VC) in Value Engineering. In order to evaluate the influence of the sample size for RS models, the random forest regression (RFR), nearest neighbor (K-NN) method, and partial least squares regression (PLSR) were conducted by combining Landsat8/OLI imagery in 2016 and 91 *Pinus densata* sample plots in Shangri-La City of China. The mean of the root mean square error (RMSE) of 200 random sampling tests was adopted as the accuracy evaluation index of the RS models and VC as a relative cost index of the modeling samples. The research results showed that: (1) the statistical values (mean, standard deviation, and coefficient of variation) for each group of samples based on 200 experiments were not significantly different from the sampling population (91 samples) by *t*-test ($p = 0.01$), and the sampling results were reliable for establishing RS models; (2) The reliable analysis on the RFR, K-NN, and PLSR models with sample groups showed that the VC decreases with increasing samples, and the decreasing trend of VC is consistent. The number of optimal samples for RFR, K-NN, and PLSR was 55, 54, and 56 based on the spherical model of variance function, respectively, and the optimal results were consistent. (3) Among the established models based on the optimal samples, the RFR model with the determination coefficient $R^2 = 0.8485$, RMSE = 12.25 Mg/hm², and the estimation accuracy P = 81.125% was better than K-NN and PLSR. Therefore, they could be used as models for estimating the aboveground biomass of *Pinus densata* in the study area. For the optimal sample size and sampling population, the RFR model of *Pinus densata* AGB was established, combining 26 variable factors in the study area. The total AGB with the optimal samples was $1.22 \times 10^7$ Mg, and the estimation result with the sampling population was $1.24 \times 10^7$ Mg based on Landsat8/OLI images. Respectively, the average AGB was 66.42 Mg/hm² and 67.51 Mg/hm², with a relative precision of 98.39%. The estimation results of the two sample groups were consistent.

**Keywords:** variance function; value coefficient; optimal sample size; aboveground forest biomass; remote sensing estimation; Landsat 8/OLI

## 1. Introduction

Forest biomass is an important indicator for estimating forest productivity, terrestrial ecosystem function, and sustainable forest management. With the fast development of remote sensing technology, multi-source remote sensing data have replaced the traditional ground sampling survey method for forest aboveground biomass inversion. It can obtain the quantity, spatial distribution, and dynamic change of forest resources and realize the quantitative inversion of the forest measurement parameters combined with var-

ious models and sample surveys [1,2]. Therefore, it can meet the requirements for monitoring and analyzing forest resources and ecological processes at different scales while saving the investigation cost. Recently, various studies have been performed to establish non-parametric remote sensing estimation models of forest aboveground biomass based on optical remote sensing data, such as decision tree regression, k-nearest neighbor method, support vector machine regression, and artificial neural network models [3–10]. The uncertainty of the model has attracted much attention in the quantitative inversion of forest biomass by remote sensing [11–14]. The above uncertainties mainly include ground data measurement uncertainty, model selection uncertainty, and spatial sampling uncertainty. Shettles et al. [15] pointed out that the model uncertainty accounted for a large proportion (about 70%) of the total uncertainty among the three uncertainties. Compared with statistical models, the number of samples used for remote sensing modeling has a significant impact on the uncertainty caused by the model parameters, and the uncertainty of model parameters gradually decreases with the increase in the number of samples [16]. The remote sensing estimation accuracy of forest biomass on a regional scale based on a statistical model relies on the model training accuracy under different sample sizes. Given traditional statistical sampling data, 30 for a small sample and 50 for a large sample are only empirical sample sizes. The larger the number of samples, the better the model's reliability. However, too many samples need to consume more human and material resources and financial resources. In forestry production, under the premise of ensuring a certain model accuracy, modeling with as few samples as possible is one of the important problems to be solved in quantitative remote sensing.

There have been few developments on the uncertainty of the sample size used for a remote sensing estimation model of forest measurement parameters at the regional scale worldwide. The main reason is to ensure the model's reliability while discussing the sample problem. However, the feature variables for the remote sensing estimation of forest measurement parameters based on statistical models have obvious mechanism problems. Moreover, the influence of the number of samples on the model accuracy depends on the modeling method. Some methods are suitable for large samples, while some can obtain superior results using small samples. Furthermore, apart from the number of samples, the samples' distribution, diversity, and representativeness are also key parameters. The effect of sample size on the analysis results is not apparent for homogeneous samples. In the previous studies based on the number of samples of the model, Fu et al. [17] analyzed the uncertainty of the estimated regional biomass based on the sample size of single-tree biomass modeling and believed that increasing the amount of modeling data can effectively improve the biomass model's estimation accuracy, accuracy, and work efficiency, and reduce the uncertainty. In practice, it is challenging to obtain forest resource survey data. Due to a limited number of samples, the model will have an "over-learning" phenomenon. The non-parametric method to estimate forest aboveground biomass can effectively solve this problem [18]. According to the estimation and application of remote sensing-based regional forest biomass, Wu demonstrated that increasing the sample size could improve the modeling accuracy, especially for the support vector machine algorithm. However, the accuracy changes reflected by partial least squares regression (PLS) and k-nearest neighbors (K-NN) algorithms indicate that increasing the number of samples cannot necessarily improve the accuracy. Thus, different estimation methods need to find the most suitable number of samples.

Aiming at the uncertainty of samples in remote sensing models of forest biomass using traditional statistical models, a new method to solve reasonable samples was suggested that should integrate the geostatistical variance function and VC in value engineering to explore the change of VC with the change of samples and then solve the reasonable sample size for a remote sensing estimation of forest biomass.

In this study, a *Pinus densata* forest, a typical forest ecosystem in Shangri-La, Yunnan Province, was taken as the research object, and the mean value of the root mean square

error (RMSE) based on 200 experiments of random sampling results as the evaluation index of model accuracy for different sample groups. Combining Landsat 8/OLI image and 91 sample plots, the comparative analysis of model accuracy was conducted to provide a feasible model of reasonable samples for forest biomass remote sensing estimation based on random forest regression (RFR), nearest neighbor value (K-NN) regression (K-NN) regression, and partial least squares regression (PLSR).

Existing and limited studies on sample optimization at a regional scale for AGB estimation are primarily concerned with how to design sample spots and select models [6,7–11]. There have been no published reports on how to model the relationships between sample sizes and estimated accuracy. Therefore, this research aims at optimising samples for statistical models and exploring the feasibility of approaches to improving forest AGB estimation accuracy in the alpine mountains of Yunnan Province, China.

## 2. Study Area and Materials

### 2.1. Description of the Study Area

The study area, Shangri-La City, is located in the northwest of Yunnan Province in southwestern China, neighboring Sichuan and Tibet (Figure 1). The geographical coordinates are 99°23'6.08"–100°18'29.15" east longitude and 26°52'11.44"–28°50'59.57" north latitude. Shangri-La has a total area of 1.142 million hm$^2$. The Jinsha River surrounds it on the east, south, and west sides. It is the junction of Yunnan, Sichuan, and Tibet provinces and the world's natural heritage "Three Parallel Rivers" scenic spot. High terrain, low heat, and low temperature are the main characteristics of Shangri-La. The altitude is 1503–5545 m, the annual average temperature is 5.5 celsius centigrade, the annual average precipitation is 618.4 mm, the average snowfall day is 35.7 d, and the annual daylight rate is 40–50%, belonging to the mountain cold temperate monsoon climate. The study area is rich in plant resources due to the dense tributaries of the Jinsha River water system, ice and snowmelt water, plateau lakes, and other water resources, and the forest soil types are dominated by brown soil and red soil. The forest vegetation area is large, the coverage rate is high, and the distribution of north–south differences is obvious. There are mainly 10 types of vegetation, including *Picea asperata*, *Abies fabri*, *Pinus densata*, *Pinus yunnanensis*, and *Quercus semicarpifolia*. *Picea asperata* in the study area is a local pioneer species and the largest area tree species. Whether natural or planted stands, they were all pure, so they were used as a case for this study.

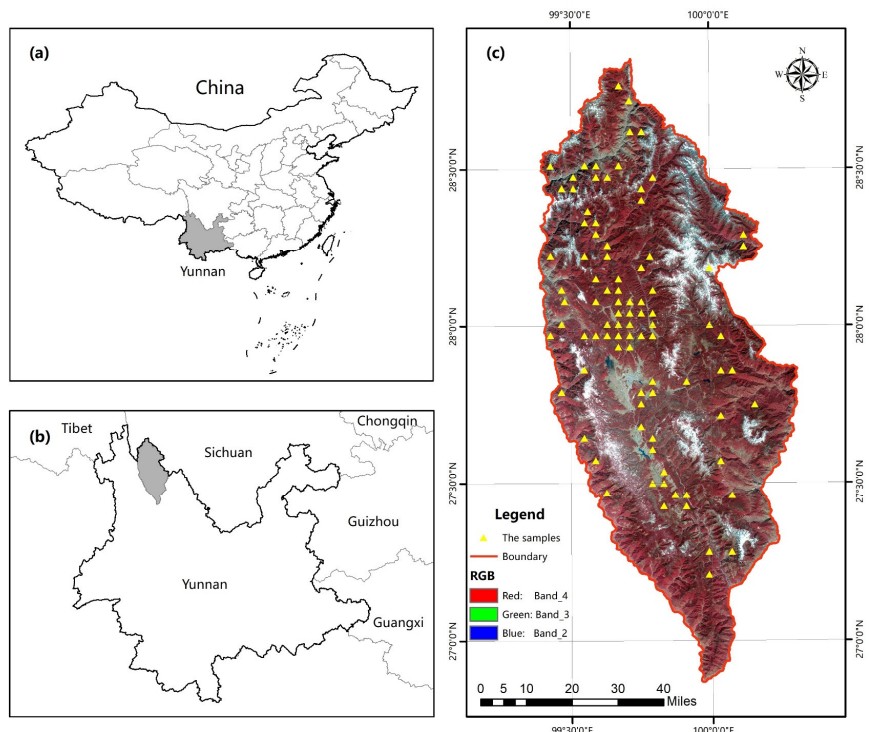

**Figure 1.** Location of study area. Shangri-La City in the northwest of Yunnan Province in south-western China. (Note: (**a**) the study area is located in Southwestern China; (**b**) Shangri-La City is part of Yunnan province, and (**c**) a standard false color composite of Landsat Thematic Mapper (TM) band 4 in red, band 3 in green, and band 2 in blue, highlighting vegetated areas in red color, yellow triangle is the 91 sample plots of *Pinus densata*).

*2.2. Sample Plots Data and Calculation of AGB*

A total of 91 sample plots in the study area were inventoried in 2016. They were circular sample plots with a size of 1 hectare, which were usually known as angle gauge controlling sample plots (AGCSP) [19]. The plots were systematically allocated on a previous spatial distribution map of forest types (Figure 1). Within each plot, the average diameter at breast height (DBH) and tree height (H) were recorded. The calculation of the AGB of each plot has two steps. Firstly, the individual-average-standard tree AGB in AGCSP was calculated by the average tree height and average diameter at the breast height of the plot. The Individual tree aboveground biomass model of *Pinus densata* is shown in Equations (1) and (2) [6,20]. Secondly, the number of trees per hectare was calculated according to Equation (3), and the sample plot's aboveground biomass was obtained by combining the single-average-standard tree aboveground biomass and the number of *Pinus densata*.

Within 91 sample plots, the minimum, maximum, mean, and the standard deviations of the aboveground biomass of different tree species are recorded in Table 1. The maximum is 133.61 Mg/hm², and the minimum is 3.36 Mg/hm².

$$AGB = 0.048(DBH^2H)^{0.880} \tag{1}$$

$$AGB = 0.0955(DBH^2H)^{0.8329} \tag{2}$$

$$N = F_g \sum_{j=1}^{k} \frac{1}{g_j} Z_j \tag{3}$$

where Equation (1) is the model of aboveground single tree biomass of *Picea asperata and Abies fabri*, and Equation (2) is the model of the aboveground single tree biomass of *Pinus*

*densata*. AGB is for aboveground biomass; DBH is the diameter at breast height; H is single tree height; $Z_j$ is the number of counted trees of the *j*th diameter interval class (assuming that there are *k* diameter classes); $g_j$ is the sectional area of the median value of the diameter class; $F_g$ is the sectional area coefficient of angle gauge; *N* is the obtained number of trees per hectare.

**Table 1.** Description of the forest biomass observations.

| Species | Sample Size (N) | Minimum (Mg/hm²) | Maximum (Mg/hm²) | Average (Mg/hm²) | SD (Mg/hm²) |
|---|---|---|---|---|---|
| *Pinus densata* | 91 | 3.36 | 133.61 | 64.56 | 31.84 |

*2.3. Collection of Remote Sensing Data and Preprocessing*

Three Landsat 8/OLI images with L1 (no radiometric calibration and atmospheric correction) products were obtained from Geospatial Data Cloud (http://www.gscloud.cn/, accessed on 13 January 2022) on 9 November 2016 (path/rows: 132/040, 132/041), while on 20 December 2016 (path/rows: 131/041). The three images adopt a Universal Transverse Mercator coordinate system (UTM) projection with zone 17 north, WGS84 ellipsoid, with a spatial resolution of 30 m × 30 m. The images have 11 spectral bands and were mosaicked into one image, while 1–7 bands (on coastal band, three visible bands, one NIR band, and two SWIR bands) were only utilized in this study. The images were pre-processed by the software ENVI 5.3, including radiometric calibration and atmospheric correction (FLAASH).

*2.4. Extraction of Feature Variables from Remote Sensing Data*

For AGB modelling at a regional scale, the variables contain spectral bands, vegetation indices, and textures [21–26]. The main feature parameters include 1–7 bands of the preprocessed image, the spectral reciprocal value of the bands, the band combination, and the texture feature factor [6]. The texture feature factor is based on eight texture feature parameters defined by Haralick et al. [1], and moving windows size (5 × 5, 7 × 7, 9 × 9 pixels) and different bands (bands 3–7) were employed to calculate the texture features.

(1)   Spectra feature parameters

For spectral bands, 23 modeling variables were extracted including the original spectral bands (b1, b2, b3, b4, b5, b6, b7), reciprocal spectral bands (1/b1, 1/b2, 1/b3, 1/b4, 1/b5, 1/b6, 1/b7), and combination bands ((b5 − b4)/(b5 + b4), b2/b5, b3/b5, b4/b5, b6/b5, b7/b5, (b4 + b6)/b7, (b4 + b6 + b7)/b5, (b3 + b4 + b6)/b7) [6].

(2)   Texture feature parameters

Textures are important features of remote sensing images, which play an important role in remote sensing image classification, quantitative remote sensing, and other fields [21,23,26]. They can represent ground object structure information in remote sensing images, reflecting the important information of spatial changes of land cover type in remote sensing images [22,24,25]. Currently, the main methods for texture feature extraction are statistical, structural, and spectral decomposition methods [6,24]. In this study, texture feature calculation was performed based on moving windows size (5 × 5, 7 × 7, 9 × 9, 11 × 11, 15 × 15, 19 × 19, 25 × 25 pixels) and Landsat8 OLI bands (band 3, 4, 5, 6, 7) according to the eight texture feature parameters defined by Haralick et al. Therefore, 280 texture feature variables were extracted as the alternative parameters of AGB estimation and modeling. The calculation equations of eight texture features are listed in Table 2.

**Table 2.** Calculation formulas of texture features.

| Texture Feature Parameters | Equations | Texture Feature Parameters | Equations |
|---|---|---|---|
| Mean, ME | $\sum_{i,j=0}^{N-1} i P_{i,j}$ | Dissimilarity, DI | $\sum_{i,j=0}^{N-1} i\, P_{i,j} \lvert i - j \rvert$ |
| Variance, VA | $\sum_{i,j=0}^{N-1} i P_{i,j}(i - ME)^2$ | Entropy, EN | $\sum_{i,j=0}^{N-1} i\, P_{i,j} \lvert -\ln P_{i,j} \rvert$ |
| Homogeneity, HO | $\sum_{i,j=0}^{N-1} i\, \dfrac{P_{i,j}}{1 + (i - j)^2}$ | Second Moment, SM | $\sum_{i,j=0}^{N-1} i P_{i,j}{}^2$ |
| Contrast, CO | $\sum_{i,j=0}^{N-1} i\, P_{i,j}(i - 1)^2$ | Correlation, CR | $\sum_{i,j=0}^{N-1} i P_{i,j}\left[ \dfrac{(i - ME)(j - ME)}{\sqrt{VA_i VA_j}} \right]$ |

Note: where $P_{i,j}$ is the probability of ($i$, $j$) appearing in the image, and $i$, $j$ are the pixel values respectively.

## 3. Methods

### 3.1. Experimental Design of Sample Groups

In general, sufficiently large training samples facilitate the construction of remote sensing models with better adaptability and stability [6–8]. However, excessive sample plots consume more human, material, and financial resources. In forestry production, it is important to design a reasonable number of samples within a model's accuracy and stability. The research was based on the number of samples and the accuracy of the model. A random sampling test was devised.

For the convenience of description, x represents the number of samples, and Z(x) represents the estimation accuracy of the models versus the number of samples. Each experiment randomly selects x samples with an interval of 2 from the sampling population (91 samples) as group samples, and the number of each group samples x (x ≥ 26) starts from 26 (due to the number of model's parameters was 26) to avoid model overfitting. The model was run, and the estimation accuracy of the model was recorded for each group. The experiment was repeated 200 times due to the difference between the mean value of 200 and 1500 trials being not significant by *t*-test ($p = 0.01$) (Figure 2). When the number of samples x exceeded the total number of samples, the experiment was over. The 200 times cycle experiment was set to avoid the randomness of a single experiment.

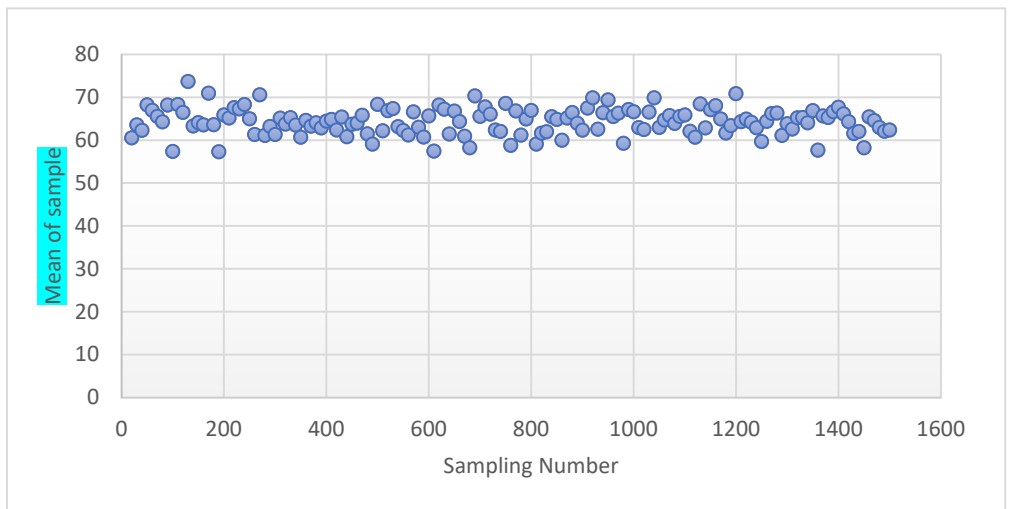

**Figure 2.** Variability of 200 experimental results (sample size = 50).

*3.2. Estimation Method and Accuracy Evaluation Indexes*

(1)   Random Forest Regression (RFR)

Random forest was proposed by Breiman and Cutler in 2001 [27]. The algorithm uses the Bootstrap sampling method. First, n random samples are taken from the original sample set as a sample set Dn. and n decision tree models Gn(x) are trained with the sample set Dn. When training the nodes of the decision tree model, an optimal feature is selected among k randomly selected sample features to do the left and right subtree division of the decision tree. This study uses a regression algorithm; then the final category is the mean of the leaf nodes reached by that sample point [28].

(2)   K-Nearest Neighbor (K-NN)

In the K-NN regression model, the observed value samples of the feature variables are designated as the reference set, the prediction set of the variables to be tested is designated as the target set, and the space defined by feature variables is the feature space. The predicted value $m_p$ of the continuous variable biomass $m$ on the pixel $p$ can be calculated as [6,20]:

$$m_p = \sum_{i=1}^{k} w_{ip} m_i \tag{4}$$

where $m_i$ is the measured value at the reference sample site $i$ for the variable $m$; $k$ is the neighbor's number when calculating the predicted value $m_p$; $w_{ip}$ is the pixel weight value calculated as follows:

$$w_{ip} = \begin{cases} d_{p_i,p}^{-t} \Big/ \sum_{j=i_1(p)}^{i_k(p)} d_{p_j,p}^{-t} & , \text{only if } i \in \{i_1(p), \dots, i_k(p)\} \\ 0, \text{other conditions} \end{cases} \tag{5}$$

where $i$ is the reference set sample; $p$ is the target set pixel; $p_j$ is the sample corresponding to the reference set sample $j$, $d_{p_i,p}^{-t}$ is the distance decomposition factor; $k$ and $t$ are constants, where their optimal values can be selected by repeated testing. $\{i_1(p), \dots, i_k(p)\}$ are the $k$ reference set samples most similar to the pixel $p$ measured in the feature space. The spatial similarity of feature variables indicated by $d_{p_i,p}$ can be obtained as:

$$d_{p_i,p} = \sqrt{\sum_{l=1}^{n_f} \left( f_{l,p_j} - f_{l,p} \right)} \tag{6}$$

where $f_{l,p_j}$ and $f_{l,p}$ are the feature variables of the spectral bands and their derivation factors of the remote sensing image corresponding to the reference set and the target set samples, respectively. $n_f$ is the number of feature variables; $p$ is the pixel of the target set; $p_i$ is the pixel corresponding to the sample $i$ of the reference set.

Referring to my previous related research results [24], the parameters of the K-NN model in this study were chosen as follows: distance metric was used in the form of Euclidean distance, with a k value equal to 8 and a t value equal to 1.

(3)   Partial Least Squares Regression (PLSR)

PLSR is a multivariate statistical analysis method that consists of a combination of multiple linear regression, principal component analysis, and typical correlation analysis. The PLSR method combines the correlation between the independent and dependent variables while extracting the characteristics, which eliminates the complex covariance of the original variables while preserving the large variance, thus allowing the created regression model to reflect the corresponding relationship between the dependent and independent variables. The PLSR method has the advantage of dealing with this problem and is able to remove unsuitable samples from the residuals of the calibration model more

easily and build the best model quickly. PLSR first constructs new variables named components, each of which is a linear combination of the dependent variable and determines its coefficients by combining the independent and dependent variables, and finally constructs the regression equation between the dependent variable and these components using the least squares method. The PLSR model is as follows [29]:

$$y_k = \lambda_{0\kappa} + \lambda_{1\kappa}T_1 + \cdots + \lambda_{n\kappa}T_n \quad (\kappa = 1, 2, \ldots, n) \tag{7}$$

where $T_1$, …, $T_n$ are the linear combinations of the bands of the spectrum, respectively, and $\lambda_i$ ($i$ =1, …, $n$) is the coefficients, which can be estimated by least squares.

(4)  Evaluation of Model Accuracy

The traditional statistical model-based accuracy evaluation indexes mainly use the coefficient of determination ($R^2$), root mean square error (RMSE), and estimation accuracy (P). Generally, the larger $R^2$, the smaller RMSE, and the higher P. RMSE (Equation (8)) were utilized to evaluate the uncertainty of the model's sample size in this study.

In order to estimate the RMSE of the models for different samples, Leave-one-out (LOO) cross-validation was employed; that is, for $N$ samples, one sample is drawn from N samples as the test set, while the remaining $N - 1$ samples were adopted as the reference set, and the cycle was repeated N times until the end. This study statistically analyzed the model predicted value $\hat{y}_i$ ($i$ = 1, …, $N$) of the $N$ samples and the measured value ($y_i$) of the corresponding sample. The model stability was evaluated using the average RMSE for each group of samples based on 200 tests.

$$\text{RMSE} = \sqrt{\frac{\sum_{i=1}^{N}(\hat{y}_i - y_i)^2}{N}} \tag{8}$$

where $y_i$ and $\hat{y}_i$ are the measured and predicted values of the sample size of the $i$th group, respectively.

*3.3. Optimal Samples Estimation Integrating Semi-Variance Functions and Value Coefficients*

(5)  Value coefficients (VC)

The uncertainty analysis was conducted on the model sample size using the value coefficient (VC), which calculating formula is VC = F/C (where $F$ is the function coefficient and $C$ is the cost coefficient). The VC means the relative ratio of the degree of matching between the function and cost of a product in Value Engineering to help engineers find engineering improvement objects and reduce costs when conducting cost analysis [29]. In this study, the relative ratios of the RMSE between per group samples and sampling population was used as the function coefficient F, and the relative ratio of the cost was used as the cost coefficient C, i.e., F = RMSE_sample/RMSE_population and C = N_sample/N_population. It indicates a relative variation between a model's accuracy (RMSE) and the cost of modeling samples based on the sampling population.

$$VC(i) = \frac{RMSE_i/RMSE_T}{N_i/N_T} \tag{9}$$

where $N_T$ is the sampling population, $RMSE_T$ is the model's root mean square error; $N_i$ is the number of $i$th group samples, and $VC(i)$ is the model's value coefficient with $N_i$ samples, $RMSE_i$ is its root mean square error.

(6)  Semi-variance Functions

Assuming that the values of the regionalized variable ($\gamma(h)$) at space points X and X + $h$ are $Z(X)$ and $Z(X + h)$, the semi-variance function, also known as the "semi-covariance function", is defined as follows [30]:

$$\gamma(h) = \frac{1}{2N(h)} \sum_{i=1}^{N(h)} [Z(x_i) - Z(x_i + h)] \tag{10}$$

where $N(h)$ is the number of point pairs at distance $h$. Since $\gamma(h)$ is unknown, it must be obtained by the relevant model from the experimental data, such as a spherical model:

$$\gamma(h) = \begin{cases} 0 & h = 0 \\ C_0 + C\left(\frac{3h}{2a} - \frac{h^3}{2a^3}\right) & 0 < h \leq a \\ C_0 + C & h > a \end{cases} \tag{11}$$

where $C_0$ is the nugget variance, $C$ is the partial sill, i.e., arch height, $C_0 + C$ is the sill, and $a$ is the range, which indicates the maximum distance of a regionalized variables from spatial autocorrelation to irrelevance.

(7)  Optimal Samples Estimation

In this study, the initial analysis was derived from scatterplots that the values of VC (*Y*-axis) were graphed against the values of cost, i.e., the number of sample plots (*X*-axis); and the approach of the parameters based on semi-variogram was suitable for the relationship between VC and the number of sample plots. Here, the change in values of VC was attributed to spatial autocorrelation, and the number of samples was regarded as the spatial distance. To estimate the optimal value of group samples, the range parameter of spatial distance was established, that is, the maximum distance of spatial autocorrelation or variability.

Based on the spherical model, $h$ is $N_i$, i.e., the number of $i$th group samples ($h = s + 26$) when solving the optimal samples. $\gamma(h)$ is the value coefficients of models, $C_o$ is the value of VC at $N = 26$ ($s = 0$), $C$ is the change rate of VC, $C_o + C$ is the maximum or minimum VC when the cost reaches its optimal sample size. When estimating the parameters of the spherical model, let $\gamma(h) = Y(x)$, $X_1 = x$, $X_2 = x^3$, $C_0 = B_0$, $\frac{3C}{2a} = B_1$, and $\frac{-C}{2a^3} = B_2$. The transformed linear model is shown in Equation (12), and the parameters ($B_0$, $B_1$, $B_2$) are obtained using the least-squares method. The optimal sample size corresponding to different models is shown in Equation (13).

$$Y(x) = B_0 + B_1 X_1 + B_2 X_2 \tag{12}$$

$$N_{optimal} = \sqrt{\frac{-B_1}{3B_2}} \tag{13}$$

## 4. Results

### 4.1. Collection of Model Feature Variables

In order to establish the remote sensing model, feature variable factors should be first selected. The correlation analysis between the aboveground biomass of *Pinus densata* and remote sensing spectral feature variables reveals a strong correlation between the spectral band combination values and aboveground biomass. There is a very significant correlation level with the variable values of (B4 + B6)/B7, (B3 + B4 + B6)/B7, B6/B5, and B7/B5 within 23 spectra feature parameters, while only 22 variables were significant within 280 texture feature parameters. This study employed 26 feature factors with highly significant correlation levels (four band combinations and 22 texture features, as shown in Figure 3) to establish AGB models. The correlation between the forest aboveground biomass and remote sensing feature variables of *Pinus densata* is shown in Figure 3. The correlation coefficients ranged from −0.24 to 0.26, with the strongest correlation being (B4 + B6)/B7. As shown in Figure 3, 7-5-CO represents contrast (CO) texture filtering under the 5 × 5 window of the 7th band, B$i$ is the $i$th band of Landsat8 OLI ($i$ = 1, 2, …, 7), and so on. All

corresponding characteristic variables of biomass estimation models for *Pinus densata*. was shown in Table 3.

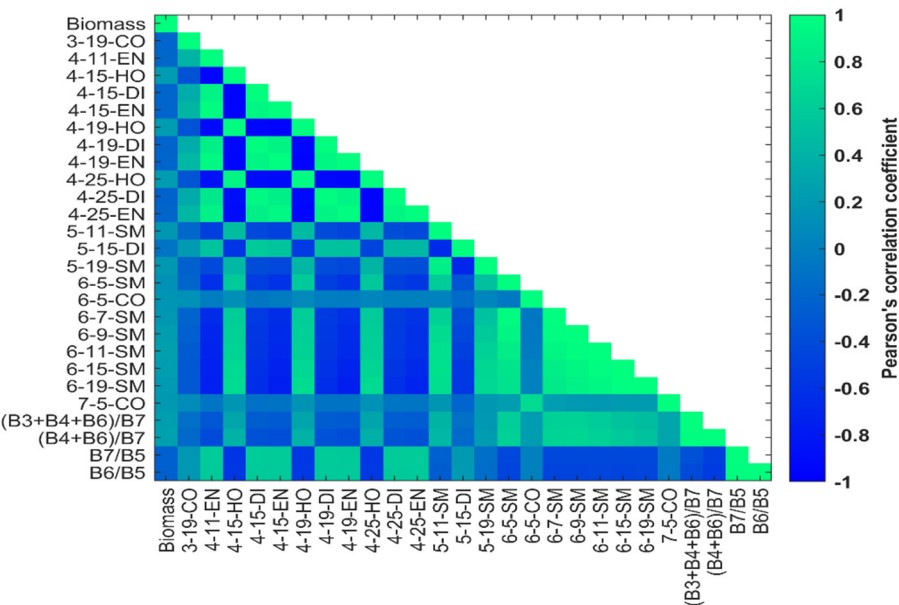

**Figure 3.** Correlation between *Pinus densata* biomass and remote sensing feature variables.

**Table 3.** Characteristic variables of biomass estimation models for *Pinus densata*.

| Variable Types | 26 Characteristic Variables |
|---|---|
| Texture features variables | 3-19-CO, 4-11-EN, 4-15-HO, 4-15-DI, 4-15-EN, 4-19-HO, 4-19-DI, 4-19-EN, 4-25-HO, 4-25-DI, 4-25-EN, 5-11-SM, 5-15-DI, 5-19-SM, 6-5-SM, 6-5-CO, 6-7-SM, 6-9-SM, 6-11-SM, 6-15-SM, 6-19-SM, 7-5-CO |
| Spectral feature variables | (B3 + B4 + B6)/B7, (B4 + B6)/B7, B7/B5, B6/B5 |

### 4.2. Sampling Effect

In general, the reliability analysis of the uncertainty of sample size on RS models accuracy should depend on sufficiently large samples. Therefore, this experiment adopted 91 sample plots of *Pinus densata* as the sampling population surveyed by the National Forest Resources Planning and Design Department in 2016 according to systematic sampling with more than 90% estimation accuracy, which can adequately represent the overall changes of typical forest ecosystems in the study area. The reliability of the sampling results requires that the mean of samples per group should be consistent with the sampling population.

Therefore, to avoid the random error of a single experiment, the samples mean of 200 random selections was used as the statistical value per group (Table 4). By *t*-test ($p = 0.01$), the statistical per group was not significantly different from the sampling population (91 samples), and the group samples could be used as the model training sample for the accuracy calculation.

**Table 4.** Statistical values of different sample sizes under 200 random sampling.

| Number (N) | MEAN (Mg/hm²) | STDV (Mg/hm²) | CV | Number (N) | MEAN (Mg/hm²) | STDV (Mg/hm²) | CV |
|---|---|---|---|---|---|---|---|
| 26 | 64.1250 | 31.1622 | 0.4885 | 60 | 64.7301 | 31.4118 | 0.4860 |
| 28 | 64.1084 | 30.9366 | 0.4852 | 62 | 64.6219 | 31.6234 | 0.4899 |
| 30 | 64.1363 | 31.3576 | 0.4916 | 64 | 64.2446 | 31.4907 | 0.4906 |
| 32 | 64.9825 | 30.9830 | 0.4792 | 66 | 64.5929 | 31.6167 | 0.4899 |

| 34 | 64.4875 | 31.5644 | 0.4920 | 68 | 64.4334 | 31.5488 | 0.4899 |
| 36 | 64.8181 | 31.1635 | 0.4829 | 70 | 64.3674 | 31.5703 | 0.4909 |
| 38 | 64.4959 | 31.3397 | 0.4876 | 72 | 64.4704 | 31.6147 | 0.4907 |
| 40 | 64.9200 | 31.5427 | 0.4873 | 74 | 64.5597 | 31.6779 | 0.4910 |
| 42 | 64.1528 | 31.3019 | 0.4893 | 76 | 64.7947 | 31.6243 | 0.4883 |
| 44 | 64.5796 | 31.3125 | 0.4859 | 78 | 64.4896 | 31.6672 | 0.4912 |
| 46 | 64.6162 | 31.4065 | 0.4871 | 80 | 64.5684 | 31.5644 | 0.4890 |
| 48 | 64.6764 | 31.4140 | 0.4868 | 82 | 64.4872 | 31.6410 | 0.4908 |
| 50 | 64.4293 | 31.7441 | 0.4938 | 84 | 64.6030 | 31.7069 | 0.4909 |
| 52 | 64.0252 | 31.2946 | 0.4896 | 86 | 64.5759 | 31.6848 | 0.4907 |
| 54 | 64.7302 | 31.2874 | 0.4841 | 88 | 64.5169 | 31.6846 | 0.4911 |
| 56 | 64.4366 | 31.4135 | 0.4883 | 90 | 64.5806 | 31.6278 | 0.4898 |
| 58 | 64.5098 | 31.6588 | 0.4914 | 91 | 64.5601 | 31.8402 | 0.4907 |

Note: MEAN is sample average of group samples, STDV is standard deviation, CV is coefficient of variation.

*4.3. Statistical Analysis of Model Accuracy*

The accuracy of the remote sensing estimation of biomass at a regional scale depends on the model, and the estimating results of different models within the same group of samples varies. In this study, three models, RFR, PLSR, and K-NN were selected to analyze the variation of model accuracy under different samples.

Table 5 shows the variation of the value coefficient of the aboveground biomass estimation model based on the RFR model. By *t*-test ($p < 0.01$), the VC difference between the group samples and the sampling population (sample size = 91) is not significant, indicating that the sampling results are consistent with the population.

In Figure 4, the VC of the three models (Figure 4a–c) decreases with the increasing cost of modeling samples. When the sample number of *Pinus densata* is less than 50, the VC variance is 1.5 times that of the sampling population, and the relative ratios of VC are large; and the sample number is larger than 50, the trend of change is flat. It indicates that increasing the sample is beneficial to the model accuracy within 50 samples; if the sample size exceeds 50, the VC change rate decreases. If the sample size is further increased, the model accuracy change should be flat. Correspondingly, the cost is too high.

**Table 5.** The value coefficients of RF model based on different sample size.

| Number (N) | VC | Number (N) | VC | Number (N) | VC | Number (N) | VC |
|---|---|---|---|---|---|---|---|
| 26 | 3.6725 | 44 | 2.1048 | 62 | 1.4970 | 80 | 1.1378 |
| 28 | 3.3303 | 46 | 2.0238 | 64 | 1.4387 | 82 | 1.1153 |
| 30 | 3.1921 | 48 | 1.9382 | 66 | 1.3992 | 84 | 1.0873 |
| 32 | 2.9421 | 50 | 1.8750 | 68 | 1.3587 | 86 | 1.0661 |
| 34 | 2.7792 | 52 | 1.7665 | 70 | 1.3244 | 88 | 1.0378 |
| 36 | 2.6112 | 54 | 1.7184 | 72 | 1.2804 | 90 | 1.0116 |
| 38 | 2.4604 | 56 | 1.6545 | 74 | 1.2512 | 91 | 1.0000 |
| 40 | 2.3421 | 58 | 1.6001 | 76 | 1.2104 | | |
| 42 | 2.2155 | 60 | 1.5333 | 78 | 1.1755 | | |

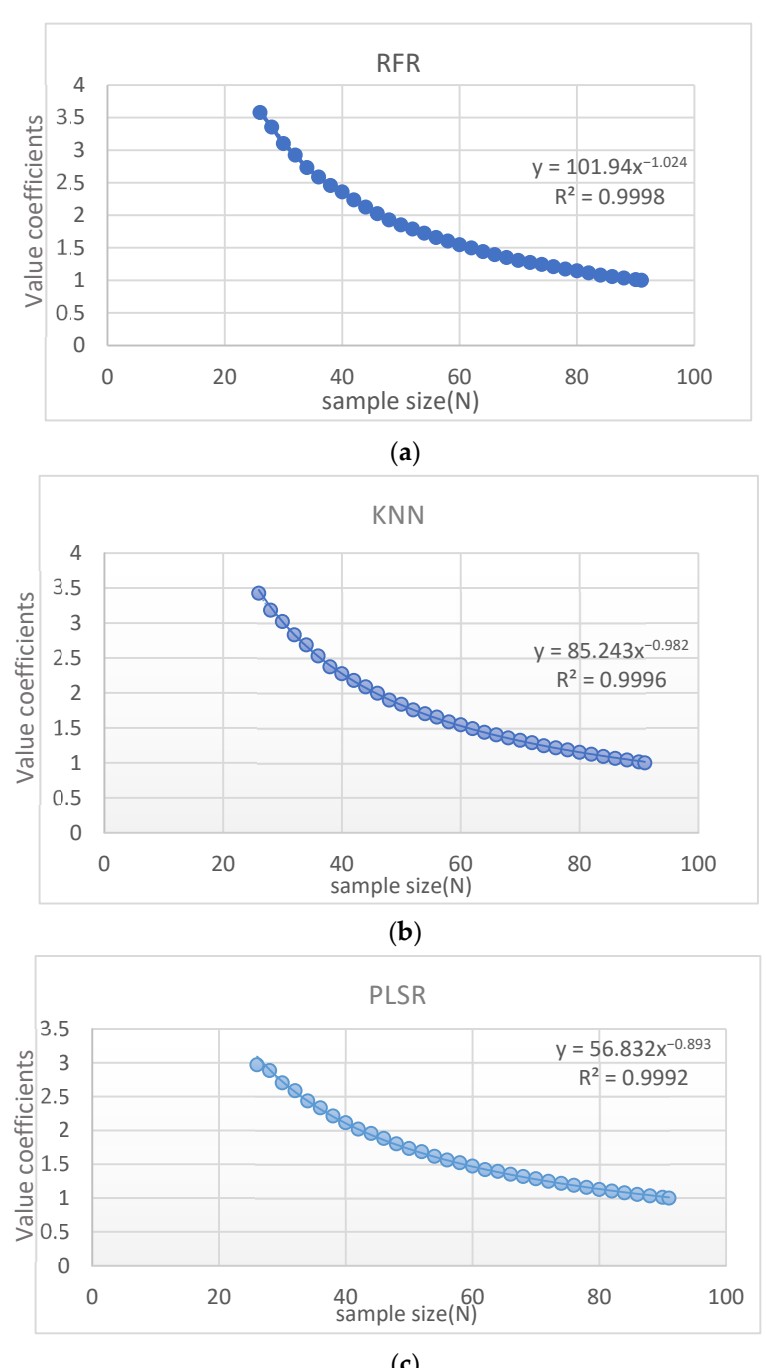

**Figure 4.** Variation of model value coefficients with sample size (**a**) RFR; (**b**) KNN; (**c**) PLSR.

### 4.4. Determination of Optimal Sample Size

Taking the VC of the models as the variance function $\gamma(h)$, and the sample size as the spatial distance $h$, the linear transformation of the spherical model was performed. Furthermore, the spherical model parameters (Nugget variance: $C_0$, Partial sill: $C$, Range: $a$, Sill: $C_0 + C$) were obtained, and the corresponding range "$a$" was the optimal number of samples. The calculation results of the variogram parameters are shown in Table 6.

The variogram results reveal that the ratio of nugget variance to sill value ($C_0/C$) reflects the degree of spatial variation of models VC to samples. Generally, it may be divided into three grades: 0–25%, 25–75%, and above 75%, indicating the weak, medium, and high spatial variation, respectively [27]. The results in Table 3 indicated that the spatial variances of samples to VC for the RFR, K-NN, and PLSR models were 61.17, 60.97, and

58.54%, all of which reached a moderate spatial variance. Correspondingly, there was a moderate spatial correlation between model accuracy and cost variation, and the spherical model parameters could be solved (Table 6). According to Equation (13), the optimal samples of RS models for the RFR, K-NN, and PLSR were 55, 54, and 56 (Table 7), and the results of the optimal sample sizes were consistent. Respectively, the modeling costs were only 60.44, 59.34, and 61.54% of the total costs (91 samples).

**Table 6.** Fitting parameters of variation function based on spherical model.

| Model | B0 | B1 | B2 | Nugget Variance $(C_0)$ | Sill $(C_0 + C)$ | Sampling Variation $(C_0/C + C_0)$ | Range $(a)$ |
|---|---|---|---|---|---|---|---|
| RFR | 3.5788 | 0.062479 | −0.000007 | 3.5788 | 5.850802 | 61.17% | 55 |
| K-NN | 3.4261 | 0.061024 | −0.000007 | 3.4261 | 5.619168 | 60.97% | 54 |
| PLSR | 2.9717 | 0.0564 | −0.00006 | 2.9717 | 5.07636 | 58.54% | 56 |

**Table 7.** Estimation accuracy with reasonable sample size.

| Model | Optimized Samples (N) | Decision Coefficient $(R^2)$ | RMSE $(Mg/hm^2)$ | Estimation Accuracy $(P\%)$ |
|---|---|---|---|---|
| RFR | 55 | 0.8485 | 12.2535 | 81.1253 |
| K-NN | 54 | 0.2658 | 28.7278 | 55.3621 |
| PLSR | 56 | 0.3972 | 28.0759 | 56.3810 |

Note: the coefficient of determination $(R^2)$, root mean square error (RMSE) and estimation accu-racy (P) were the mean of estimation accuracy with optimized samples based on 200 tests.

### 4.5. Forest AGB Estimation Based on Optimized Samples

For the optimal sample size, the RFR, K-NN, and PLSR models of AGB *Pinus densata* were established in the study area, combining 26 variable factors. The modeling accuracy is shown in Table 7. The results showed that the mean of estimation accuracy, coefficient of determination $(R^2)$, root mean square error (RMSE), and estimation accuracy (P) were 0.85, 12.2535 $Mg/hm^2$ and 81.13% for the RFR model, which is better than K-NN ($R^2$ = 0.27, RMSE = 28.7278 $Mg/hm^2$, P = 55.36%) and PLSR ($R^2$ = 0.40, RMSE = 28.0759 $Mg/hm^2$, P = 56.38%) based on 200 tests. Therefore, the RFR model was used to estimate the AGB of *Pinus densata* in the study area.

Based on the sub-compartment data of the National Forest Resources Planning and Design Survey in the study area, the distribution area of *Pinus densata* was extracted, with a total area of 183,671.1470 $hm^2$. Using the RFR model and Leave-one-out cross-validation, combined with the optimal sample size of 55 and the sampling population (91 samples), the coefficients of determination of the modeling accuracy were 0.852 and 0.8478 (Figure 5a,b), which was better than the K-NN ($R^2$ = 0.0464, Figure 5c; $R^2$ = 0.2078, Figure 5d) and PLSR ($R^2$ = 0.0548, Figure 5e; $R^2$ = 0.1501, Figure 5f) models, and the total AGB was 1.22 × 10$^7$ Mg and 1.24 × 10$^7$ Mg based on Landsat8/OLI images, and the average AGB was 66.42 $Mg/hm^2$ and 67.51 $Mg/hm^2$, respectively, and their spatial distribution is shown in Figure 6a,b.

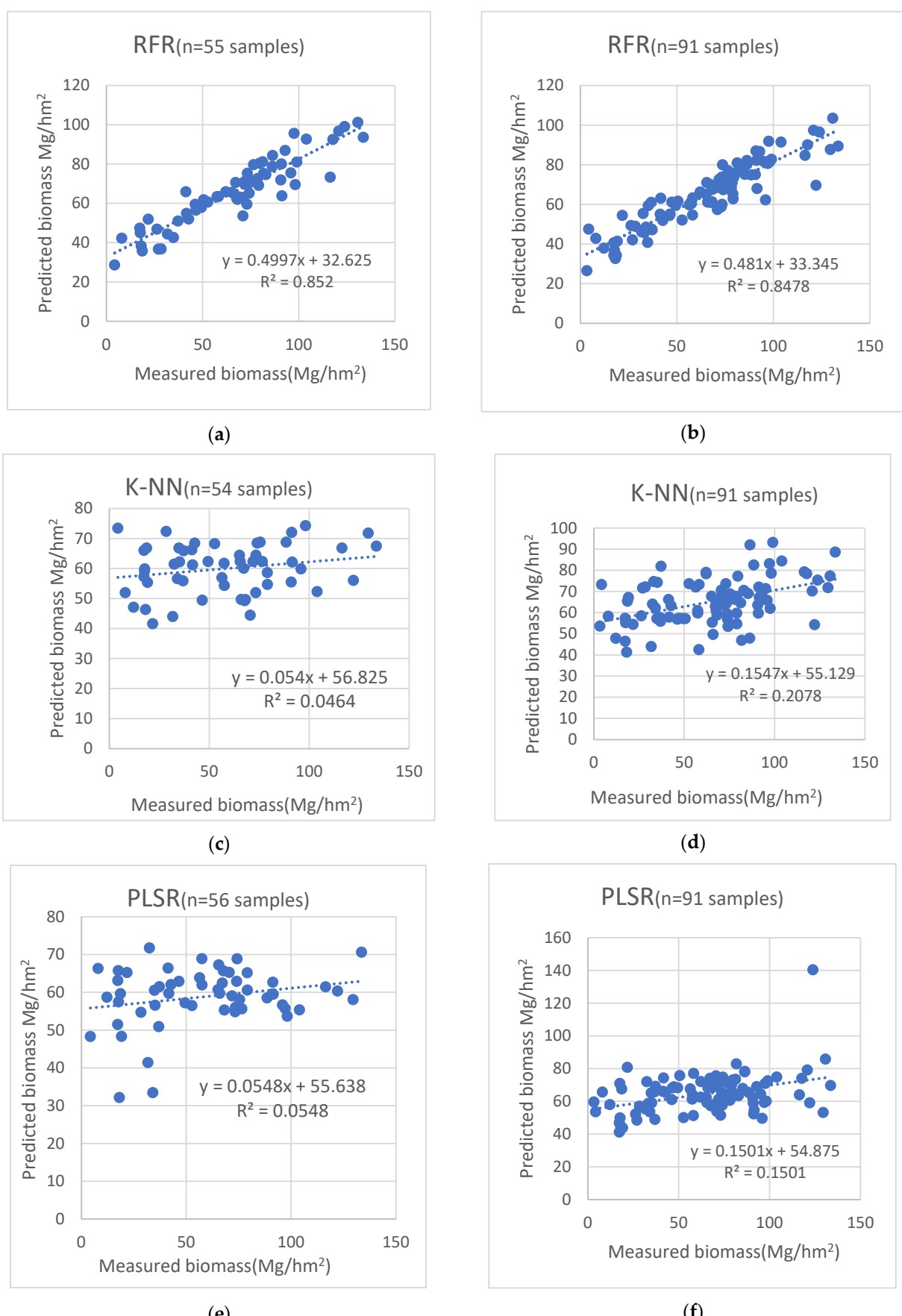

**Figure 5.** Biomass measured value compared with RFR (**a**,**b**), K-NN (**c**,**d**), and PLSR (**e**,**f**) model predicted value.

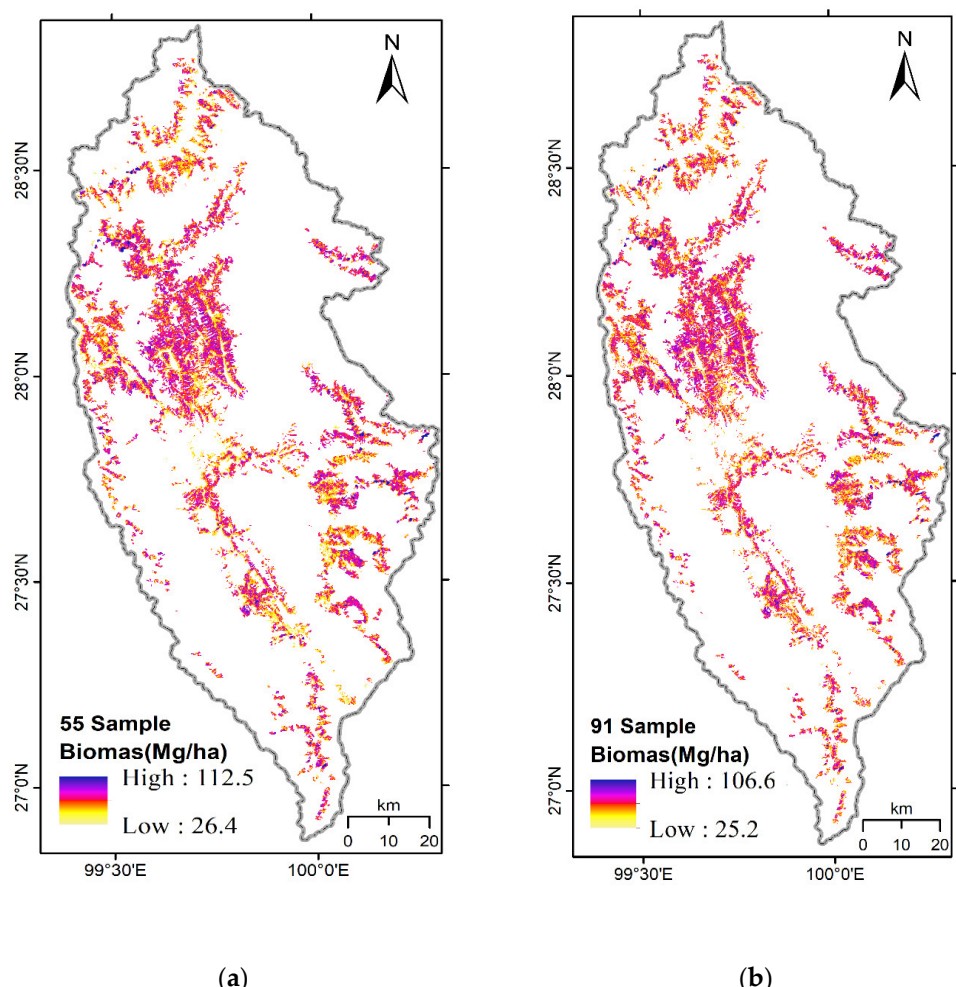

**Figure 6.** Map of biomass inversion basic data (**a**) Inverse map of alpine biomass from 55 samples. (**b**) Inverse map of biomass from 91 samples.

## 5. Discussion

### 5.1. Sample Size Problem for Remote Sensing Estimation of Forest Aboveground Biomass at Country Scale

In this paper, we estimated the optimal samples for RS models, such as the random forest regression (RFR), nearest neighbor (K-NN) method, and partial least squares regression (PLSR) by combining Landsat8/OLI imagery and 91 sample plots at a regional scale. The optimal results of the RFR, K-NN, and PLSR models were 55, 54, and 56, which were solved by the least squares method based on linear transformation to the spherical model of variance function. The idea behind these models is that as the value coefficient of the model's RMSE decreases, the number of sample plots, i.e., modeling cost, change quickly at the beginning, then slowly and eventually smoothly. When the value coefficient becomes smooth, the corresponding numbers of the modeling samples (cost of models) can be regarded as optimal samples. The optimal results of the three models were consistent.

While the available research on the remote sensing estimation of forest AGB was based on the traditional empirical sample size, i.e., 30 is a small sample and 50 is a large sample [7,9,11,13,14]. In general, the larger the number of samples, the better the model's reliability. There are few reports on the uncertainty of the sample size of the remote sensing estimation model at the regional scale. This is mainly due to the fact that the optimal sample size changes with the object, such as *Pinus densata, Pinus yunnanensis*. On the other

hand, it requires not only a stable model and a sufficiently large sample size but also a normal distribution of samples and a stable evaluation index of model accuracy variation. In this study, the optimal sample size was solved by integrating geostatistical variance function and value coefficient in value engineering, which was reconstructed using the model accuracy evaluation index RMSE and the model sample cost. The optimizing method of the sample size is one of the innovative points of this study.

### 5.2. Selection Problem for Remote Sensing Estimation Models of AGB and Feature Variables

The nonparametric models, such as neural network models (NNM), support vector machine regression (SVMR), RFR, K-NN, and parametric models such as linear regression analysis, PLSR, are often used to develop AGB estimation at region scale [6–12,21–26]. When analyzing sample uncertainty based on remote sensing estimation models, firstly, the models of RS should be reliable; secondly, the influence of sample size on model accuracy is related to the estimation methods, some of which are suitable for large samples such as K-NN, and some of which can obtain better results with small samples such as SVMR. In this study, the nonparametric models RFR, K-NN, and parametric PLSR models were selected because the RFR model is better than the others in anti-noise and voiding the risk of overestimations by introducing random factors [27]; the K-NN model is suitable for large samples because of non-assumptions on the data and non-sensitivity to abnormal samples [28], and the PLSR model can effectively eliminate the model parameter covariance problem [31].

According to the results of the selected three modes (RFR, K-NN, and PLSR), it seems that the optimal sample sizes are almost the same (55, 54, and 56), although the RMSEs for them are significantly different (12.25 vs. 28.73 or 28.08). It is implied that the optimization results have little relationship with the estimation accuracy of the selected model for aboveground biomass.

To further verify whether the optimal method was correlated with the selected model, an experiment using the SVMR model was conducted. The optimal method was similar to that of using the RFR model and cross-validation. By combining Landsat8/OLI imagery and 91 sample plots at region scale, the optimal number of samples for the SVMR model was 54, for which the coefficients of determination of model and estimation accuracy were 0.011 and 51.49%, respectively. The optimization result of the SVMR model was almost the same as those of the three selected models (RFR, K-NN, and PLSR). It shows that the optimal samples are independent of the accuracy of the selected model. The optimal results of the selected models were consistent.

Zhao's study [21] has implied that the potential uncertainty of remote sensing estimation models may be caused by optical imagers with different temporal, spatial, radiometric, and spectral resolutions. The feature variables values extracted from optical images were different even at the same time, in the same place, and for the same feature object. In this study, 26 independent variables with textural features and vegetation indexes were selected by correlation analysis between AGB and variables feature based on Landsat8 OLI. The difference in optimizing samples was not significant by modeling with 26 independent variables. For other optimal images, such as SPOT5, IKNOS, QuickBird, and MODIS, the result should be analyzed in future work.

### 5.3. Validity of Estimation Results Based on Optimal Sample Size

The verification of the accuracy of the aboveground biomass estimation results of forests at a regional scale has been a difficult problem in quantitative remote sensing [21–26,31–36]. The main reason is the difficulty of obtaining the field measurement values at the regional level. The existing studies on the validity of estimation results at a regional scale on AGB estimation focus mainly on how to improve remote sensing models' accuracy evaluation, such as optimizing model algorithms [6], optimizing model features variables [21,34], and using multi-source remote sensing collaboration [12,33–36].

In Lu's study [12], aboveground forest biomass was estimated with Landsat and Li-DAR data. Although the estimation accuracy of the remote sensing model was high, the validity of the estimation result depended on the model itself and sample spots. As the validity is lacking a sampling control statistical survey in the study area, the credibility of the results needs to be further improved.

In this study, the aboveground biomass of *Pinus densata* was counted using the data measured by the National Forest Resources Planning and Design Survey in 2016 in the study area. Wang's [32] study showed that the total biomass of *Pinus densata* was $1.3347 \times 10^7$ Mg, and the average biomass was 72.6705 Mg/hm². Using Wang's results as the real reference value, the estimated results based on the RFR regression model were $1.22 \times 10^7$ Mg and $1.24 \times 10^7$ Mg at the sample sizes of 55 and 91, with absolute precision of 91.41 and 92.90%, respectively. This indicates the reliability of the RFR model based on the optimal sample size.

### 5.4. Optimal Solution Problem about Equations (12) and (13)

Wang's [32] study shows that Equations (12) and (13) have three cases after calculating $B_0$, $B_1$, and $B_2$.

(1) $B_0 > 0$, $B_1 > 0$, $B_2 < 0$, when the three parameters (the nugget variance $C_0$, the partial sill $C$, and the range a) are optimally fitted based on spherical model of variation functions, Equations (12) and (13) have optimal solutions.

(2) $B_0 < 0$, $B_1 > 0$, $B_2 < 0$, as $B_0 < 0$, that is, the parameter $C_0 < 0$, it does not meet the requirements of the spherical model. So it is necessary to let $B_0 = 0$, then the Equation (12) becomes $Y(X) = B_1X_1 + B_0X_2$, and Equation (13) has the optimal solution.

(3) $B_0 > 0$, $B_1 > 0$, $B_2 \geq 0$, if $B_2 = 0$, Equation (13) becomes $Y(X) = B_0 + B_1X$. For a linear model, not a spherical model, the parameters can be solved according to the estimation method of the parameters of the linear regression model. The other is $B_2 > 0$ when the original data are adjusted by adding or deleting some unimportant data points from the actual variance function points and repeatedly adjusting it many times until $B_2 < 0$.

## 6. Conclusions

In the paper, integrating the theory of semi-variance function in geostatistics and value coefficients in value engineering, a new method suggested that a reasonable sample size was estimated by remote sensing models of forest biomass. The main conclusions are as follows:

(1) The statistical values (mean, standard deviation, and coefficient of variation) for each group of samples based on 200 experiments are not significantly different from the overall samples (91 samples) by *t*-test ($p = 0.01$), and the sampling results were reliable for establishing RS models.

(2) The reliable analysis of value coefficients based on RFR, K-NN, and PLSR models with sample groups shows that the VC decreases with increasing samples of every group, and the decreasing trend of VC is consistent. The optimal samples of RFR, K-NN, and PLSR were 55, 54, and 56 based on the spherical model of variance function, respectively, and the optimal results are consistent.

(3) Among the established models based on the optimal samples, the RFR model with the determination coefficient $R^2 = 0.8485$, RMSE = 12.25 Mg/hm², and the estimation accuracy P = 81.1253% was better than K-NN and PLSR. It could be used as a model for estimating the aboveground biomass of *Pinus densata* in study area. Based on the optimal 55 samples of the RFR model and overall (91 samples), the total aboveground biomass in the study area was $1.22 \times 10^7$ Mg and $1.24 \times 10^7$ Mg, and the average aboveground biomass was 66.42 Mg/hm² and 67.51 Mg/hm², respectively, with a relative precision of 98.39%, and the estimation results of two groups were consistent.

**Author Contributions:** Q.S. designed the experiments and analyzed the data and wrote the paper; Y.P. discussed the method and reviewed the paper; K.W. wrote the manuscript; F.X. and L.X. performed the experiments and modelling; H.S. conducted remote sensing data and calculation of aboveground biomass at the plot level and region scale. All authors have read and agreed to the published version of the manuscript.

**Funding:** This research was funded by the National Natural Science Foundation of China (Grant nos. 31860205 and 31460194) and the Yunnan Provincial Education Department Scientific Research Fund Project (No. 2021Y249), China, in 2021.

**Conflicts of Interest:** The authors declare no conflict of interest. The funding sponsors had no role in the design of the experiment; the collection and analyses of data; the writing of the manuscript.

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
