# Peer review of "Optimization of Samples for Remote Sensing Estimation of Forest Aboveground Biomass at the Regional Scale"

_remotesensing, doi:10.3390/rs14174187_

Round 1

Reviewer 1 Report

1. Introudction

Line 99: “In the paper, the Pinus densata forest, a typical forest…” THE paper? Wich paper? I suggest changing this to In this work or In this study or In this analysis. In this paragraf

2. Study area and materials

To include the coordinates is necessary but since Remote Sensing is an international journal, I think it´s a good idea for the authors to include country and region/continent, not just the province. Personally, I don´t know what country Yunnan Province is in.

Line 186. 2.5. Collection of Model Feature Variables. I suggest that this item be moved to the results section.

4. Results

Line 365: I suggest replacing “…and the corresponding range a was the optimal number..” with “…and the corresponding range “a” was the optimal number” for readability..

Author Response

Dear Reviewer:

    First of all, thank you very much for your valuable comments on this study, which will be of great help to this study. We will respond to all your comments point-by-point after discussion. Please see the attachment for detailed responses.

Reviewer 2 Report

Comment for remote sensing-1766227

This manuscript addressed optimization of samples for remote sensing estimation of forest AGB at county level. Whole text is easy to read and to follow. I reviewed it and provided some special comments as follows.

1.      Title point out “county level”. Audiences might not clearly understand why “county level” is so important in remote sensing field. I suggest that authors could have more descriptions in Introduction chapter.

2.      Line 34-35 “Based on the optimal 55 samples of RFR model and sampling population (91 samples), the total AGB in the study area was 1.22×107 Mg and 1.24×107 Mg, and the average above-ground biomass was 66.42 Mg/hm2 and 67.51 Mg/hm2.” The sentence is not easy to understand because audiences might confuse that the sample number is 55 or 91? Therefore, authors might improve it.

3.      I suggest adding the study purpose in the end of introduction.

4.      In Study Area and Materials, authors should explain the difference of equations (1) and (2) used in this study.

5.      All samples (91) are Pinus densata?  If so, this conifer should be consider to put in Title. The importance of this conifer also should be emphasized in Introduction.

6.      Figure 4 should be improved because X and Y axis should have title.

7.      Figure 5 also should be improved because X and Y axis should add unit.

8.      Discussion chapter is too simple, authors should find more relevant papers to discuss here (such as 5.3 section)

Overall, I feel that this study is interesting in remote sensing field. Therefore, I am pleased to recommend it for publication in the remote sensing.

Author Response

(The authors gave the same response as above.)

Reviewer 3 Report

This manuscript attempted to utilize the variance function and value coefficient for optimal sample size selection, which is essential for aboveground biomass estimation since it is difficult to collect a large number of samples, and this work will help guide sample collection via fieldwork. The major issues in this manuscript are as follows.

(1).  The optimal sample size varies with the model selected for aboveground biomass estimation. According to the results of the selected three modes (RFR, k-NN, PLSR), it seems that the optimal sample sizes are almost the same (55, 54, and 56). However, the RMSEs for them are significantly different (12.25 V.S. 28.73 or 28.08). Whether it is the optimal sample size for k-NN or PLSR cannot be drawn without the comparison experiment with that with more samples (such as 91 samples). As a result, it is suggested to give the results of accuracy assessment similar to that using the RFR model and cross-validation (see line 389).

(2).  Besides, more models with different requirements of samples (such as SVMR) should be given. It should not be examined in future research because the optimal sample sizes for the three models in this manuscript are equivalent, which cannot indicate the optimizing method is valid for different models.

Finally, some minor problems should be resolved.

(3).  In figure 4, the axis labels should be given, and the ‘sample size’ in the figure title should also be specified.

(4).  In figure 3, there is a spelling mistake of ‘meam of sample’ in the y-axis title.

(5).  In line 207, the sentence is with a grammatical mistake.

(6).  What does the ‘sampling effection’ mean? It is uncommon to use the word of ‘effection’.

Author Response

(The authors gave the same response as above.)

Round 2

Reviewer 3 Report

Major revision

The authors DO NOT resolve the questions raised in the last version. Specifically,

1.       The optimized samples of 54 for k-NN using CV model DO NOT mean it is the optimal sample number for k-NN if results derived from different sample size is not given.  The intermediate result of 54 samples as the optimized option should be validated. Otherwise, the comparison with RFR does not make scene. The similar problem can be found in the result of optimized samples using PLSR. Please add this part.

2.       Assumed that the optimized samples for k-NN and PLSR are true after validation, the results of optimized samples (55, 54, and 56) show that the models (RFR, k-NN, PLSR) have the similar requirements of samples. The comparison within the similar models in sample demands is NOT convinced. Thus, it is suggested to add an experiment with a different model, such as SVM.

Author Response

Dear Reviewer:

    Once again, thank you for your valuable comments, which were very helpful for this study. For all your suggestions, we have responded to your queries one by one after discussion. Cross-validation scatter plots and SVMR model experiments have been added. Please see the attachment for detailed responses.

This manuscript is a resubmission of an earlier submission. The following is a list of the peer review reports and author responses from that submission.